# A Powerful LAMP Weapon against the Threat of the Quarantine Plant Pathogen *Curtobacterium flaccumfaciens* pv. *flaccumfaciens*

**DOI:** 10.3390/microorganisms8111705

**Published:** 2020-10-31

**Authors:** Stefania Tegli, Carola Biancalani, Aleksandr N. Ignatov, Ebrahim Osdaghi

**Affiliations:** 1Laboratorio di Patologia Vegetale Molecolare, Dipartimento di Scienze e Tecnologie Agrarie, Alimentari Ambientali e Forestali (DAGRI), Università degli Studi di Firenze, Via della Lastruccia 10, 50019 Sesto Fiorentino (Firenze), Italy; carolabiancalani@virgilio.it; 2Agrarian and Technological Institute (ATI), Peoples’ Friendship University of Russia, Miklukho-Maklaya str.8, 117198 Moscow, Russia; an.ignatov@gmail.com; 3Department of Plant Protection, University of Tehran, Karaj 31587-77871, Iran; eosdaghi@ut.ac.ir

**Keywords:** Loop-Mediated Isothermal Amplification, LAMP, *Curtobacterium flaccumfaciens* pv.* flaccumfaciens*, bacterial wilt of bean, tan spot of soybean, quarantine plant pathogen, molecular diagnostics

## Abstract

*Curtobacterium flaccumfaciens* pv. *flaccumfaciens* (*Cff*) is a Gram-positive phytopathogenic bacterium attacking leguminous crops and causing systemic diseases such as the bacterial wilt of beans and bacterial spot of soybeans. Since the early 20th century, *Cff* is reported to be present in North America, where it still causes high economic losses. Currently, *Cff* is an emerging plant pathogen, rapidly spreading worldwide and occurring in many bean-producing countries. Infected seeds are the main dissemination pathway for *Cff*, both over short and long distances. *Cff* remains viable in the seeds for long times, even in field conditions. According to the most recent EU legislation, *Cff* is included among the quarantine pests not known to occur in the Union territory, and for which the phytosanitary inspection consists mainly of the visual examination of imported bean seeds. The seedborne nature of *Cff* combined with the globalization of trades urgently call for the implementation of a highly specific diagnostic test for *Cff*, to be routinely and easily used at the official ports of entry and into the fields. This paper reports the development of a LAMP (Loop-Mediated Isothermal Amplification) specific for *Cff*, that allows the detection of *Cff* in infected seeds, both by fluorescence and visual monitoring, after 30 min of reaction and with a detection limit at around 4 fg/μL of pure *Cff* genomic DNA.

## 1. Introduction

*Curtobacterium flaccumfaciens* pv. *flaccumfaciens* (*Cff*) is a highly damaging plant pathogenic Gram-positive bacterium, causing serious diseases among several cultivated and wild leguminous species. The host range of this pathogen includes common bean (*Phaseolus vulgaris* L.), cowpea (*Vigna unguiculata* L.), mung bean (*V. radiata* L.), and soybean (*Glycine max* L.) [1,2]. *Cff* was firstly described in 1922 in the USA and was later recorded in Canada, Mexico, South America, Australia and, more recently, in Iran [1,3,4]. *Cff* was listed into the A2 list of the European and Mediterranean Plant Protection Organization (EPPO) for pests recommended for quarantine regulation since 1975 [5], and since 2019 it is included in the List 1, Annex II A, of quarantine pathogens for EU that have not known to occur in the Union territory [6,7]. Currently, in Europe *Cff* is reported to be present in Russia and Turkey, although with a restricted distribution [5,8]. However, recent sporadic reports have been recorded in Spain, Germany, and Austria, with disease outbreaks occurring on bean, soybean, and cowpea, respectively, which have been promptly followed by *Cff* eradication [5,9,10]. These outbreaks raised great concern among the EU Member States, for which the dry pulse crop area has grown considerably since 2013 as a result of the current Common Agricultural Policy (CAP) towards a more sustainable agriculture [11], and where the climatic conditions are favorable to the establishment of *Cff* [2]. This quarantine phytopathogen is also a potential threat to the wide biodiversity of the many bean cultivars and other legumes originating from Europe. Most of these local varieties are promoted worldwide as traditional products. They are marked by specific European food quality logos, and often represent an essential income in rural areas.

In this overall frame, the seedborne nature of *Cff* is a serious concern as it allows the rapid spread of this plant pathogen over long distances, through the movement of infected seeds on a global market scale [12,13,14]. Although most *Cff*-infected seeds are asymptomatic, the current testing methods are still based on the visual examination of seed crops and harvested seeds [1]. Some semi-selective culture media and immunological tests have also been reported for *Cff*, but they generally have low sensitivity and specificity, and are time-consuming [for a recent review, 4]. Currently, among the PCR-based assays developed so far for *Cff* [15,16], the primer pair *Cff*FOR2/*Cff*REV4 has been demonstrated to be highly effective and sensitive for the detection of *Cff* in contaminated bean seeds [16,17,18], and in/on alternative host plants and weeds [19,20,21,22].

As no effective chemicals against this pathogenic bacterium are known, the availability of a specific and sensitive diagnostic test for the rapid detection of *Cff* on plant materials, possibly at the port of entry and into the fields, is a pivotal step towards the successful prevention of disease outbreaks. Traditional PCR-based assays are mostly unsuitable for direct field use, as well as for less well-equipped laboratories. Conversely, Loop-Mediated Isothermal Amplification (LAMP) is a technique particularly useful for on-site testing [23]. The LAMP reaction is isothermal, and thus can be performed in a simple heating block or a water bath, without the need for any specialized thermal cycler. In addition, positive results can be observed in less than an hour with the naked eye or by using a portable fluorescent reader, thus removing the post-amplification steps of electrophoresis, gel staining, and imaging. Recently, several LAMP assays have been successfully developed for some plant pathogenic bacteria [24,25,26,27,28], and sometimes also tested at field level [24,27].

This paper describes the development of a simple, sensitive, and highly specific LAMP assay for the cost- and time-saving quali-quantitative detection of *Cff*, whose effectiveness was successfully compared to the conventional PCR-based method adopted so far for this phytopathogen. The LAMP-specific primers here designed are targeting the *Cff* genomic region specifically amplified with the primer pair *Cff*FOR2/*Cff*REV4 on any *C. flaccumfaciens* isolate pathogenic on bean, regardless of its host of isolation. This LAMP assay was shown to be highly effective in detecting *Cff* in infected bean plants and seeds.

According to its performances, this LAMP method is a valuable diagnostic tool, which can be easily applied where an early and rapid diagnosis of the presence of this quarantine phytopathogen on bean seeds is required.

## 2. Materials and Methods

### 2.1. Bacterial Strains and Growth Conditions

The bacterial strains used in this study are listed in Table 1. The *Cff* strains include several different colony variants of this pathogen, that are yellow-, red- and orange-pigmented strains [21]. In addition to *Cff* strains isolated from leguminous plants and from several alternative hosts, *C. flaccumfaciens* isolates that are not pathogenic on beans, and the type strains of the four other *C. flaccumfaciens* pathovars *betae*, *ilicis*, *oortii*, and *poinsettiae*, were also used. Furthermore, closely related species (i.e., *Clavibacter michiganensis* subsp. *michiganensis* (*Cmm*)), as well as other important bean phytopathogens (i.e., *Pseudomonas savastanoi* pv. *phaseolicola* (*Psp*) and *Xanthomonas axonopodis* pv. *phaseoli* (*Xap*)), were also tested (Table 1).

Bacterial strains were routinely grown at 26 °C on Luria Bertani (LB) [29] medium both as liquid and solid cultures, while *Cff* strains were specifically plated on nutrient broth yeast extract agar medium (NBY) [30]. Bacteria were preserved for long-term storage at −80 °C in LB broth, supplemented with 40% glycerol (*w*/*v*), and subcultured when required.

### 2.2. Bacterial DNA Extraction and Thermal Lysis

Bacterial DNA extraction and purification were carried out using the Puregene^®^ DNA Isolation Kit (Qiagen GmbH, Hilden, Germany), as recommended by the manufacturer. The yield and quality of the extracted DNA were evaluated both spectrophotometrically by using Nanodrop ND-100 (Nanodrop Technologies, Waltham, MA, USA), and visually by standard agarose gel electrophoresis (1% agarose (*w*/*v*) in TBE 1X) [31], respectively. The DNA was then stored at −20 °C until needed. Bacterial DNA was also obtained by thermal lysis of single colonies, each picked up from fresh agar plates with a sterile loop and diluted in sterile bidistilled water (100 µL/pellet), incubated at 95 °C for 15 min, and then immediately cooled on ice. After a quick spin in a microcentrifuge, 5 µL lysate was directly used in amplification reactions as a template.

### 2.3. Primer Design

Several primer sets for the specific LAMP amplification of *Cff* DNA were designed by using the Primer Explorer V5 software (Eiken Chemicals, Tokyo, Japan) (http://primerexplorer.jp/lampv5e), targeting the same 306 bp nucleotide sequence on which the conventional PCR-based protocol used for *Cff* identification was also based [16,17]. A manual check for the best default program parameters and a BLAST search for potential aspecific homologies (http://www.ncbi.nlm.nih.gov/blast) were also carried out. For comparison, the primer pair *Cff*FOR2/*Cff*REV4 was also used [16]. Primers were synthesized and HPLC-purified at Eurofins (Hamburg, Germany).

### 2.4. LAMP Reaction

The 25 μL LAMP reaction mixture contained 15 µL of GspSSD Isothermal Mastermix ISO-001 (Optigene, Horsham, UK), the *Cff*F3 and *Cff*B3 outer primers (200 nM each), the *Cff*FIP and *Cff*BIP inner primers (800 nM each), and, as a template, a 5 µL solution of purified genomic DNA from *Cff* strains reported in Table 1, at a concentration depending on the experimental purposes. In negative controls, the template consisted of sterilized molecular grade bidistilled water. Cross-reactivity with non-target species was tested, by using purified DNA from non-pathogenic *C. flaccumfaciens* strains, or closely related bacteria (i.e., *Cmm*), or other plant pathogenic bacteria occurring on beans (i.e., *Psp* and *Xap*). The optimized reaction was run at 63 °C for 30 min in the CFX96™ real-time fluorometer (Bio-Rad, Hercules, CA, USA), with fluorescence data recorded at 60 s time intervals. The LAMP generated amplicons (5 μL/reaction) were also endpoint analyzed on agarose gel (2% agarose (*w*/*v*) in TBE 1X) [31], with 5 μL reaction/well, then stained with ethidium bromide (0.5 µg/mL), and UV-visualized by using the Molecular Imager^®^ Gel Doc™ XR System (Bio-Rad).

A colorimetric LAMP assay was also performed, by using the WarmStart Colorimetric LAMP 2× master mix (New England BioLabs, Ipswich, MA, USA), according to the manufacturer’s recommendations, and carried out at 63 °C for 30 min in a 25 μL final reaction. Sterilized molecular grade bidistilled water was included in each analysis as a negative control. Six independent experiments were conducted with three replicates for each sample.

### 2.5. Detection of Cff in Artificially and Naturally Infected Bean Plant Materials

The *in planta* performances of the newly developed *Cff*-specific LAMP assay have been assessed both on bean plants artificially inoculated with *Cff* and on naturally *Cff*-infected bean seeds of the Iranian cultivar (cv.) Pak.

Artificial inoculations on certified commercial bean seeds of the Italian cv. Cannellino (Linea Mediterranea srl, Pomezia-Roma, Italy) were performed by using the hilum injury method [32]. Each seed was injured by piercing the hilum with a sterile needle. Then, bean seeds were soaked for 1 h in a fresh *Cff* bacterial suspension (OD_600_ = 0.5, corresponding approximately to 10^8^ colony-forming unit/mL (CFU/mL)). Seeds used as negative controls were soaked in sterile physiological solution (SPS; 0.85% NaCl in bidistilled water). Inoculated and uninoculated bean seeds were then separately sown in 15 cm diameter plastic pots (1 seed/pot) with Cornell peat-lite mix, and incubated in a growth chamber with a 16 h photoperiod and temperature of 22 ± 2 °C. Plants were monitored daily for the appearance of symptoms, and at 14 days post-infection (d.p.i.), three true leaves from each *Cff*-inoculated and control plant were detached and used for DNA extraction, which was carried out with NucleoSpin^®^ Plant II (Macherey-Nagel, Düren, Germany), according to the manufacturer’s recommendations. The resulting total genomic material contained both the host plant (bean) DNA and the bacterial DNA (*Cff*). This DNA was then used as a template for both conventional PCR and LAMP assays. Three independent experiments were performed, with six replicates for each experimental condition for each run.

Samples (10 seeds each, corresponding to about 4.225 ± 0.350 g) of healthy-looking and symptomatic *Cff*-naturally infected bean seeds cv. Pak were separately washed three times in sterile distilled water. These seeds were derived from a *Cff*-infested area in Iran. *Cff*-free certified seeds of the same cultivar were used as a negative control. Each sample was then finely grounded, and transferred into 20 mL of SPS, under shaking conditions for 12–14 h. Any seed particle and residue were then eliminated by filtering the suspension on a sterile gauze, and the filtrate was then centrifugated at 6000× *g*, for 20 min at 4 °C. The supernatant was discharged, while the pellet was resuspended in 1 mL SPS, and then directly used for the amplification tests. Three independent experiments were performed, and six samples for each condition (i.e., healthy-looking and symptomatic *Cff*-infected seeds, and certified *Cff* free seeds) at each run were used.

## 3. Results

### 3.1. Design and Selection of LAMP Primers

As a target sequence for LAMP primer design, the highly conserved 306 pb DNA fragment amplified by the *Cff* specific endpoint PCR-based test [4,16,17] (GenBank Accession Numbers AJ307048, AJ307049, and AJ307051) was selected. Among the primers generated by Primer Explorer V5, a set of four LAMP primers was chosen as the most appropriate candidate, according to several key parameters, such as GC content, melting temperature (Tm), distances between primers, and stability of primer ends expressed as free energy (ΔG). These primers were also *in silico* analyzed by BLAST searches, and no homology hits were found but the expected *Cff* target sequence (data not shown).

The selected LAMP primers recognize six distinct regions on the target sequence, and consist of the two outer primers named *Cff*F3 and *Cff*B3, and the two inner hybrid primers named *Cff*FIP and *Cff*BIP (*Cff*FIP = sequences F1c + F2; *Cff*BIP = sequences B1c + B2). The primer sequences and their main features are reported in Table 2. Their annealing sites on the target sequence are shown in Appendix A. The stability of the ends of the selected LAMP primers, expressed as ΔG, appeared to be high, in particular to those ends essential as starting points for DNA synthesis. In particular, the 3′ ends of the outer primers *Cff*F3 and *Cff*B3, and of the sequences F2 and B2 of the internal hybrid primers, showed ΔG values definitely lower than −4 Kcal/mol (Table 2), to guarantee a high degree of stability [33]. Similarly, the ΔG values for the 5′ ends of the F1c and B1c sequences of the internal primer pair were −5.17 and −5.69, respectively.

Finally, primer design is the main challenging step in the development of a new LAMP assay as the probability of secondary structures formation is increased by the high number of primers required (four as a minimum). According to their ΔG values, the LAMP primers here designed and selected were predicted to be poorly prone towards the formation of both cross- and self-dimers as well as of hairpins (Appendix A).

No additional suitable loop primers were identified using Primer Explorer V5, with the short size of the template as the only limiting factor. However, loop primers have been demonstrated to be not essential for a successful LAMP reaction, although a reduction in amplification times can sometimes be achieved [34,35,36].

### 3.2. Optimization of LAMP Assay for Cff Detection

The optimal temperature and reaction time of the LAMP assay for *Cff* detection have been established by using pure DNA of the *Cff* type strain ICMP 2584 (40 ng/reaction) as template, and accordingly to the assessment of both gel electrophoresis and real-time fluorescence data.

Pure *Cff* ICMP 2584^T^ DNA was tested at six different amplification temperatures, within the range 62–67 °C, and the reaction time was 30 min as recommended by the manufacturer. The typical ladder-like DNA multiple bands of LAMP reaction have been detected on an agarose gel in positive samples for all the tested amplification temperatures (Figure 1A).

No unspecific amplifications or primers cross-annealing were observed, regardless of the reaction temperature. Similarly, the real-time fluorescence monitoring showed successful reactions to take place within the range 62 °C–67 °C on positive samples, and the plateau was reached after 15 min (Figure 1B). The reaction threshold times ranged from 5 to 9 min for amplification temperatures between 62 °C and 65 °C, while higher times (i.e., 12 and 15 min) were needed at higher temperatures (i.e., 66 °C–67 °C) (Figure 1). Accordingly, the optimal conditions established for this *Cff*-specific LAMP assay were 63 °C with a 30-min running time (corresponding to 30 cycles of 1 min/each), then used for all the following applications.

### 3.3. Specificity and Sensitivity of the LAMP Assay for Cff Detection

The specificity of the LAMP primers designed here was assessed by testing, as a template, the purified genomic DNA (40 ng/reaction) from 35 *Cff* strains, having different geographical origin and isolated from several different host plants, including *P. vulgaris* L., *V. unguiculata* L., *Capsicum annum* L., *Solanum lycopersicum* L., and *S. melongena* L. Moreover, DNAs from non-target phytopathogenic bacteria were also included, such as several *C. flaccumfaciens* strains, or the type strains from the *C. flaccumfaciens* pathovars *betae*, *ilicis*, *ooorti*, and *poinsettiae*, or bacteria taxonomically related to *Cff* (i.e., *Cmm*), or bacteria that are pathogenic on common bean and seedborne (i.e., *Psp* and *Xap*) (Table 1). All the *Cff* strains tested here gave positive results, as assessed by both real-time fluorescence monitoring and gel electrophoresis analysis. No cross-reactivity or aspecific amplification were found to occur when DNA from non-target bacteria species was used as a template (Table 1). Therefore, this LAMP assay for *Cff* detection has a specificity fully comparable with that of the existing conventional PCR test for this quarantine phytopathogen [16,17] (Table 1).

In order to evaluate the analytical sensitivity of the LAMP assay developed here, the smallest known amount of target DNA detected in each test sample was assessed. Several ten-fold serial dilutions of *Cff* ICMP 2584^T^ pure genomic DNA were used as a template, prepared in sterilized molecular grade bidistilled water and ranging from 100 fg/reaction to 100 ng/reaction, in a 25 µL final volume. The fluorescence-based real-time monitoring showed that this LAMP reaction is able to detect *Cff* genomic DNA down to 0.1 pg/reaction after 20 min. As expected, the amplification time was shorter as the DNA template concentration increased, and positive signals were obtained just after 15 min with 1 pg/reaction *Cff* DNA as a template (Figure 2A). The agarose electrophoresis analysis of LAMP products confirmed these data. The characteristic ladder-like DNA bands were observed when *Cff* DNA was used as a template in the range of 100 ng–100 fg per reaction (Figure 2B).

### 3.4. LAMP Detection of Cff on Artificially and Naturally Infected Bean Samples

Firstly, bean plants cv. “Cannellino” artificially inoculated with *Cff* ICMP 2584^T^ by hilum injured method were used to validate the LAMP assay for *Cff* detection. Initially, the DNA extracted from both *Cff* inoculated and uninoculated bean plants were subjected to conventional PCR, carried out using the primer pair *Cff*FOR2-*Cff*REV4 [16,17]. Positive results were obtained only for *Cff* inoculated plant samples, where the characteristic single 306 bp amplicon was specifically visualized on agarose gel (Figure 3A).

When the LAMP reaction was monitored by fluorescent real-time analysis, the DNA extracted from every artificially *Cff*-inoculated plant showed a positive amplification curve, just after 5 min. No amplification signals were recorded using, as a template, the DNA extracted from control uninoculated plants (Figure 3B). Data were further confirmed by agarose gel electrophoresis analysis of the LAMP products (data not shown).

This LAMP assay has been evaluated also on naturally *Cff*-infected bean seeds, where the presence of this phytopathogen is often asymptomatic. To this aim, bean seeds belonging to the cv. “Pak”, collected in Iran from fields previously assessed as *Cff*-infected, were used for DNA extraction [17], and then tested by conventional PCR with the primer pair *Cff*FOR2/*Cff*REV4 and by LAMP fluorescence real-time monitoring (data not shown). The infection level was about 72%. Seeds to be considered as *Cff* infected were those testing positive both by PCR and LAMP assays, regardless of the presence of any symptoms.

A protocol for the direct visual detection of LAMP results was also developed. The pH indicator dye used here turned from pink to yellow, as a consequence of the pH value decrease upon DNA amplification [37]. This protocol for visual LAMP was applied and tested on naturally *Cff*-infected bean seeds belonging to the cv. “Pak”, and highly reproducible results were obtained. As shown in Figure 4, the colorimetric visual readouts of LAMP results were positive within a time period of 15 to 30 min after the start of the incubation at 63 °C, when DNA extracted from the *Cff*-naturally infected bean seeds cv. “Pak” was used as a template, regardless of whether or not these seeds showed or not any symptoms.

The detection rate was similar to that previously obtained by conventional PCR and LAMP fluorescence monitoring. Samples containing, as a template, DNA from *Cff*-free certified seeds cv. “Pak” always scored as negative. Controls with DNA template replaced with sterile PCR-grade water also scored as negative. At last, no false positive or false negative results were detected.

## 4. Discussion

*Cff* is an important seedborne bacterial phytopathogen, with a high negative impact on the yields of beans as well as of other legumes [2,4,38,39]. The increasing worldwide spreading of *Cff* in bean-producing areas is strongly driven by the global expansion of international trades of plant materials, as well as by the multifaceted impact of climate change on plant-pathogen interaction and on weather conducive conditions [40]. It is important to point out that leguminous crops are essential for food security, as they are a staple food and help fighting hunger in less developed countries, and contribute to healthy eating habits worldwide. Furthermore, legumes have a pivotal role in a sustainable agriculture, and in climate change mitigation. Their symbiotic relationship with nitrogen-fixing rhizobia bacteria allows these crops to strongly improve soil health, and to reduce the expensive and polluting use of synthetic nitrogen fertilizers in agriculture [41].

Therefore, more research is urgently needed on these crops, as well as on their most important diseases and pathogens, including *Cff*. As it occurs in other seedborne phytopathogens, the availability of accurate and highly specific seed health tests is among the most important and essential means of control for *Cff.* Indeed, dramatic outbreaks can easily be caused by inadequate surveillance and quarantine regulatory procedures, following the entry and establishment of a new and emerging plant pathogen. Indeed, the interception of *Cff* during a port of entry inspection of plant material is extremely difficult, because *Cff*-infected seeds are mostly asymptomatic [1,38]. Furthermore, the existing assay developed so far for *Cff* detection on plant material [16,17] requires several specialized equipment and instruments and takes longer times in comparison to other most recent molecular diagnostic approaches such as LAMP [23]. LAMP has also several advantages in comparison to traditional PCR-based approaches, such as its low cost, ease in use, and suitability for in-field/on-site testing.

The only challenging step of the LAMP approach is the correct design of effective primers. Due to the high number of primers needed for each primer set, and the limited options as far as target sequences are concerned, the risk related to the formation of secondary structures can be high. In this paper, we present the development of a LAMP assay for the specific detection of *Cff*. The newly developed assay targets the unique and highly conserved 306 bp sequence, that was successfully used for the design of the PCR-based assay for *Cff* and widely applied for *in planta* testing [16,17].

The optimized conditions established for the *Cff*–specific LAMP detection are 63 °C as amplification temperature, and 30 min as running time. The analytical sensitivity of the LAMP assay for *Cff* detection is 100 fg/reaction, which was detected by real-time fluorescence monitoring after 20 min, with 5 min needed to detect 100 ng/reaction. The sensitivity was further confirmed by tests carried out on *Cff*-infected plant samples. The four LAMP primer set here designed was thus demonstrated to be highly performant, without loop primers, that sometimes have been reported to cause unwanted instability in amplification [42,43].

Lastly, a procedure was developed to visualize LAMP amplification easily and directly through naked eyes, 30 min after isothermal incubation of samples. The colorimetric pH-sensitive indicator used here allowed the avoidance of the usual drawbacks related to other colorimetric indicators, often producing false-positive results [44,45,46]. To the best of our knowledge, this is the first time that a LAMP assay was developed for the specific quali-quantitative detection of *Cff*. This LAMP assay provides a reliable, specific, and sensitive testing procedure, able to reveal the presence of *Cff* DNA on bean seeds and other plant materials, rapidly and with the naked eye, thus substantially reducing the risk of disease outbreaks by *Cff* due to its early detection.

## Figures and Tables

**Figure 1 microorganisms-08-01705-f001:**
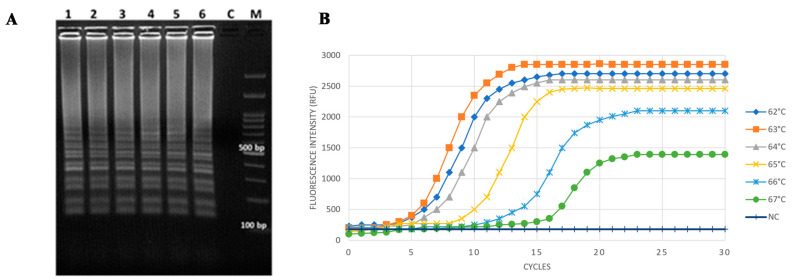
Optimization of LAMP reaction for *Cff* DNA detection. Isothermal amplification was carried out on *Cff* ICMP 2584^T^ DNA (40 ng/reaction) used as a template, and at temperatures ranging from 62 °C to 67 °C. Data assessment was based on 2% agarose gel electrophoresis analysis (**A**) and real-time fluorescence monitoring (**B**). (**A**) M - Gene Ruler 100 pb Plus DNA Ladder; C = negative control; lanes 1 to 6: amplicons in the temperature range 62 °C–67 °C. (**B**) Optimization of reaction time, according to real-time fluorescence amplification plots; NC = negative control. 1 cycle = 1 min reaction.

**Figure 2 microorganisms-08-01705-f002:**
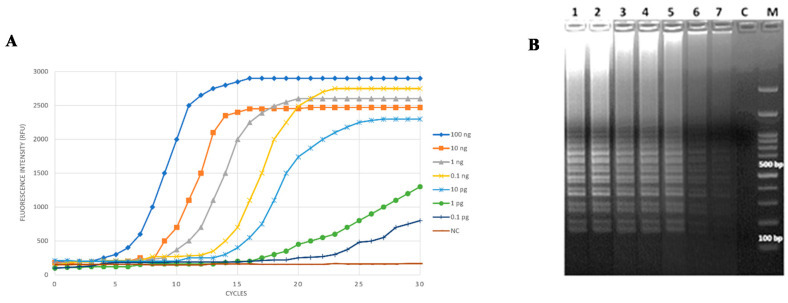
Determination of the analytical sensitivity of the LAMP assay for *Cff* detection. Isothermal amplifications were carried out on tenfold diluted test samples of *Cff* ICMP 2584^T^ DNA (from 100 ng to 100 fg per 25 µL reaction). Data assessment was based on (**A**) real-time fluorescence monitoring and (**B**) 2% agarose gel electrophoresis analysis of the LAMP products. (**A**) NC = negative control. 1 cycle = 1 min reaction. (**B**) Lanes 1–7: 100 ng, 10 ng, 1 ng, 0.1 ng, 10 pg, 1 pg, and 0.1 pg of genomic DNA/reaction, respectively; M - Gene Ruler 100 pb Plus DNA Ladder; C = negative control.

**Figure 3 microorganisms-08-01705-f003:**
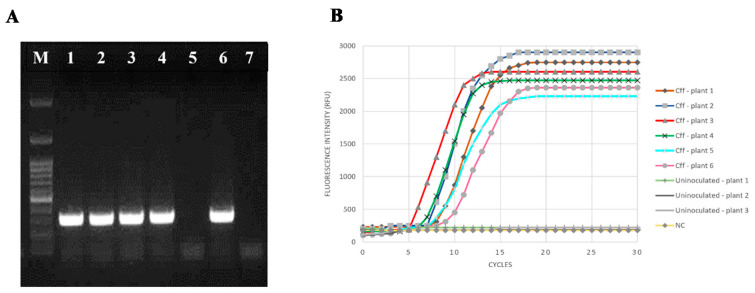
LAMP assay results from *Cff*-artificially infected bean plants cv. Cannellino. The *Cff* type strain ICMP 2584 was used for artificial inoculation. (**A**) *Cff* infection was confirmed by conventional PCR carried out using the primer pair *Cff*FOR2/*Cff*REV4. Lanes: 1, 2, 3, 4: *Cff*-inoculated bean plants; 5: uninoculated bean plant; 6: *Cff* ICMP 2584^T^ pure DNA; 7: negative control, sterilized molecular grade water as a template; M - Gene Ruler 100 pb Plus DNA Ladder. (**B**) Real-time monitoring of LAMP reaction for *Cff* detection on bean plants artificially inoculated with *Cff* ICMP 2584^T^ or with SPS. Sterilized molecular grade water was used as a negative control (NC).

**Figure 4 microorganisms-08-01705-f004:**
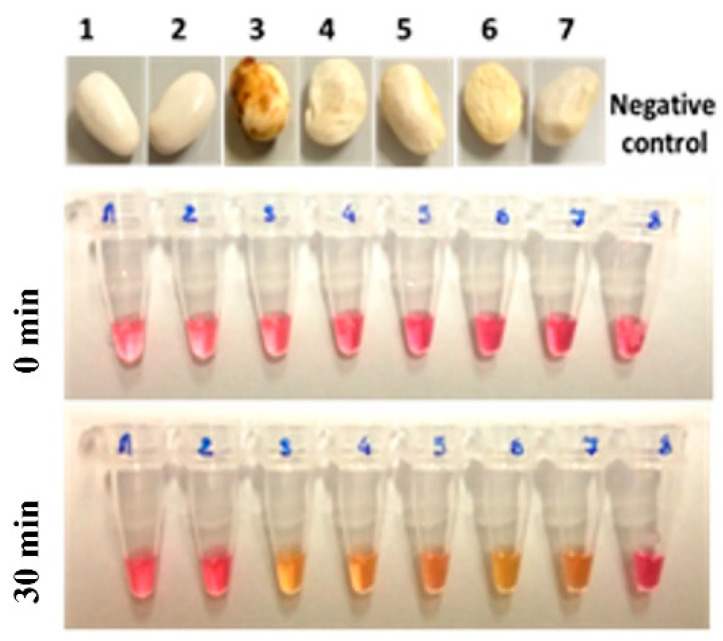
Visual detection of LAMP products obtained with bean seed samples cv. “Pak” for *Cff* detection. Visual detection of LAMP products from bean seed samples cv. “Pak” for *Cff* detection. Positive samples show a color change, from pink (negative samples) to yellow. Uninfected (samples 1 and 2) and naturally *Cff*-infected (samples 3, 4, 5, 6, 7). Molecular grade water was used as a template in negative controls (sample 8).

**Table 1 microorganisms-08-01705-t001:** Bacterial strains used in this study.

Bacteria	Strain	Source ^a^	Host of Isolation	Pathogenicity ^b^	LAMP ^c^	PCR ^c^
*Curtobacterium flaccumfaciens* pv. *flaccumfaciens* (*Cff*)	Type	ICMP 2584	Common bean	+	+	+
	50R	ICMP 22071	Common bean	+	+	+
	80O	ICMP 22069	Common bean	+	+	+
	Cb222	ICMP 21399	Common bean	+	+	+
	Cb302	−	Common bean	+	+	+
	Cb926	−	Common bean	+	+	+
	Cff110	−	Common bean	+	+	+
	Cff137	ICMP 22066	Common bean	+	+	+
	Cff151	−	Common bean	+	+	+
	Cff153	−	Common bean	+	+	+
	Cff155	−	Common bean	+	+	+
	Cff156	−	Common bean	+	+	+
	Cff178	NCPPB 178	Common bean	+	+	+
	Cff558	NCPPB 558	Common bean	+	+	+
	Cff567	NCCPB 567	Common bean	+	+	+
	Cff1412	NCPPB 1412	Common bean	+	+	+
	Cff1751	NCPPB 1751	Common bean	+	+	+
	10eg	ICMP 22079	Eggplant	+	+	+
	Cw101	−	Cowpea	+	+	+
	Cw110	−	Cowpea	+	+	+
	Eg502	ICMP 22055	Eggplant	+	+	+
	Eg505	ICMP 22054	Eggplant	+	+	+
	P701	ICMP 22078	Bell pepper	+	+	+
	P99O	ICMP 22053	Bell pepper	+	+	+
	Tom50	ICMP 22062	Tomato	+	+	+
	Tom803	ICMP 22083	Tomato	+	+	+
	Tom806	ICMP 22059	Tomato	+	+	+
	Tom930	ICMP 22057	Tomato	+	+	+
	Tom999	ICMP 22082	Tomato	+	+	+
	CFgs5	−	Soybean	+	+	+
	CFgs6	−	Soybean	+	+	+
	CFgs12	−	Soybean	+	+	+
	CFgs14	−	Soybean	+	+	+
	CFgs15	−	Soybean	+	+	+
	CFgs18	−	Soybean	+	+	+
*Curtobacterium flaccumfaciens*	Tom827	ICMP 22084	Tomato	−	−	−
	Xeu15	ICMP 21400	Chilli pepper	−	−	−
	Cmmeg20	ICMP 22056	Eggplant	−	−	−
	CFha4	−	Sunflower	−	−	−
	CFha5	−	Sunflower	−	−	−
	CFha8	−	Sunflower	−	−	−
*Curtobacterium flaccumfaciens* pv. *betae* (*Cfb*)	Type	ICMP 2594	Red beet	−	−	−
*Curtobacterium flaccumfaciens* pv. *ilicis* (*Cfi*)	Type	ICMP 2608	American holly	−	−	−
*Curtobacterium flaccumfaciens* pv. *oortii* (*Cfo*)	Type	ICMP 2632	Tulip	−	−	−
*Curtobacterium flaccumfaciens* pv. *poinsettiae* (*Cfp*)	Type	ICMP 2566	Poinsettia	−	−	−
*Clavibacter michiganensis* subsp. *michiganensis* (*Cmm*)	Type	ICMP 2550	Tomato	−	−	−
*Pseudomonas savastanoi* pv. *phaseolicola* (*Pph*)	Type	NCPPB 1449	*Lablab purpureus*	−	−	−
*Xanthomonas axonopodis* bbpv. *phaseoli* (*Xap*)	Type	NCPPB 3035	Common bean	−	−	−

^a^ ICMP International Collection of Micro-organisms from Plant, Auckland, New Zealand; NCCPB National Collection of Plant Pathogenic Bacteria, Sand Hutton, UK. ^b^ Pathogenicity assessed on tests carried out on common bean plants. ^c^ Samples testing positive (‘+’) or negative (‘−’), using LAMP assay and endpoint PCR with *Cff*FOR2/*Cff*REV4 primer pair [16,17].

**Table 2 microorganisms-08-01705-t002:** Sequences and features of the LAMP primers designed and used in this study for *Cff* specific detection.

Primers Name	Type	Primer Sequence 5′-3′	Length (bp)	Tm (°C)	GC (%)	5’ ΔG (kcal/mol)	3’ ΔG (kcal/mol)
CffF3	Forward outer	CGTTAGTGAAGGCTGACGAA	20	59.3	50	−4.51	−5.26
CffB3	Reverse outer	TTCCCGGTGTTCAGTTGAC	19	59.2	53	−6.14	−4.67
*Cff*FIP (F1c + F2)	Forward inner	GTTTGCATCCGTACGGGGCG-ACTAGCACCGACGGAACC	38 (20 + 18)	65.8; 60.6	65; 61	−5.17; −4.18	−7.97; −5.46
*Cff*BIP (B1c + B2)	Reverse inner	TTCGGTCCTGCAGTTAGTCAGC-GAATAGTTCGCGGCGTGG	40 (22 + 18)	64.2; 60.2	55; 61	−5.69; −3.08	−5.75; −7.18

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
