# Peer review of "A Powerful LAMP Weapon against the Threat of the Quarantine Plant Pathogen Curtobacterium flaccumfaciens pv. flaccumfaciens"

_microorganisms, 2020, doi:10.3390/microorganisms8111705_

Round 1
Reviewer 1 Report
The manuscript describes the development of a new lamp test to detect CFF in plant materials. The technical part of the development of the LAMP test(primer design, optimisation, sensitivity and specificity) is clear, well explained and results are convincing. However, the origin of the plant extracts used to develop the LAMP is not clear (done by the authors? If yes, where are the infected plants from, when were the plants sampled and stored and how did they extract the bact DNA? Provided by another lab? If yes, which ones, and how did they extracted the Cff DNA? (If this was already mentioned in other articles, mentioning the references would be all right). While the positive controls are clear, the negative controls are not well described and a few of them are missing (such as water with LAMP reagent only and materials from other plant species).
The discussion needs to be re-organised. All the ideas are present, but it should start with the results of the lamp and why those are important (fast field-adapted detection, part of fighting climate change and disease expansion…) and finished with the findings of the manuscript and the “request” for more research on these types of crops.
The text (especially introduction and discussion) needs some English revisions. I have suggested some edits and changes that the authors might choose to accept or not. After corrections, I strongly suggest to ask an English-speaking person to proof-read..
Suggestions for English edits and questions (underlined)
Line 20-21: “…over short and long distances. Cff remains viable in the seeds for long times, even in field conditions.”
Line 23 & 24: Remove “so far” and “Conversely”.
Line 27-29: This paper reports the development of a LAMP (Loop‐Mediated Isothermal Amplification) specific for Cff, that allows the detection of Cff in infected seeds, both by fluorescence and visual monitoring, after 30 min of reaction and with a detection limit at around 4 fg/μl of pure Cff genomic DNA.
Line 36: Replace “affecting” by “among”
Line 37: remove “mainly”
Line 39: Replace “and later it was also recorded” by “and was later recorded”
Line 40-43: Cff was listed into the A2 list of the European and Mediterranean Plant Protection Organization (EPPO) for pests recommended for quarantine regulation since 1975 [7], and is included into the List 1, Annex II A, of quarantine pathogens for EU that are not known to occur in the Union territory since 2019 [5,6].
Line 44: remove “just”
Line 44-45: “However, recent sporadic….”
Line 48-51: These outbreaks raised great concern among the EU Member States, for which the dry pulse crop area has grown considerably since 2013 as a result of the current Common Agricultural Policy (CAP) towards a more sustainable agriculture [11], and where the climatic conditions are favorable to the establishment of Cff [2].
Line 52: “…bean cultivars and other legumes originated from Europe”
Line 53: How are the varieties protected ?, Also, I am not sure that the variety/cultivar protection is relevant to the manuscript. I would have written “These autochthonous (maybe replace autochthonous by local?) varieties are promoted worldwide as traditional products and marked by specific European food quality logos, and are therefore an essential income in rural areas”.
Line 56-59: The seedborne nature of Cff is a serious concern as it allows rapid spread over long distances, through movement of infected seeds on a global market scale.
Line 58-60: Despite the asymptomatic aspect of the infected seeds, the current testing methods are still based on the visual examination of seed crops and harvested seeds.
Line 61: Replace “have been also reported” by “have also been”
Line 62-63: Replace “…specificity, besides to be time‐consuming such as occurring usually for the most traditional diagnostic methods” by “…specificity and are time consuming”
Line 64: Replace “have been demonstrated highly effective” by “ have been demonstrated to be highly effective”; Remove “also”.
Line 65: Replace “into contaminated bean seeds [16‐18], as well as on alternative host plants and..” by “in contaminated bean seeds [16‐18], alternative host plants and weeds.” The issue of colour variant is not clear. How is it related to the PCR testing?
Line 67: Replace “Since no effective chemicals against this damaging plant pathogenic bacterium are known yet, ..” by “As no effective chemicals against this pathogenic bacterium are known, …”
Line 73: Replace “thus it can be performed” by “ thus can be performed” and “or even in a water bath” by “or a water bath”
Line 75-76: Replace “by naked eye or by a portable fluorescent reader, removing the post‐amplification steps given by of electrophoresis, gel staining and imaging.” By “with the naked eye or by using a portable fluorescent reader, thus removing the post‐amplification steps of electrophoresis, gel staining and imaging.”
Line 76-77: Replace “Several LAMP assays have been already successfully developed for some plant pathogenic bacteria even in recent times [24‐28].” By “Recently, several LAMP assays have been successfully developed for some plant pathogenic bacteria [24‐28].”. Can you mentioned if some of them were used directly in the field?
Line 83: Replace “shown highly effective to” by “shown to be highly effective to”. Remove “also”. Most of the introduction is about the importance of testing the seeds to detect infection at the port of entry, and in the last paragraph, the authors mentioned that the LAMP is “also” efficient on seeds. The authors should place the emphasis on the seed testing, while using plant materials to develop the test.
Line 90-91: This sentence is not clear. What are the other pathovars in the table 1?
Line 113: remove “respectively”.
Line 122: Remove “available so far”.
Line 143: Replace “the Cff‐specific LAMP assay here developed” by “the newly developed Cff‐specific LAMP assay “.
Line 147: Remove “respectively”
In the materials and methods (line 143-159), it is not clear which seeds are artificially inoculated. In line 146-147, it is mentioned that “healthy and Cff‐naturally infected seeds of the Iranian bean cv. Pak” were used, but the line 148+ describes the artificial inoculation of the Italian cultivar (cv.) Cannellino. Also, how do the authors know that the seeds are healthy : are the seeds coming from registered breeder or areas with no disease?
Line 160: How many samples?
Line 193-195: sentence not clear, please rephrase.
Line 199-200: Replace “not essential” by “to be not essential”
Line 209: Replace “on agarose gel at all the amplification temperatures here tested in positive samples” by “on agarose gel in positive samples at all the tested amplification temperatures”.
Line 210: Remove “ever”.
Line 229: Replace “ here designed” by “here designed”.
Line 236: Replace “here tested” by “tested here”, and “both by” by “by both”.
Line 238: Replace “no target” by “non-target”
Line 241: Replace “here developed” by “developed here”
Line 245: Replace “showed this LAMP reaction able to detect Cff” by “showed that this LAMP reaction is able to detect Cff…”
Line 248-251: sentence not clear, please rephrase.
Section 2.1. and Line 230-234: It is not clear if the extraction of Cff and other bact were done by the authors or if the authors used plant extracts that were stored or provided by other labs. Please clarify.
Line 279: Replace “Accordingly, when LAMP reaction was monitored…” by “ When the LAMP reaction was monitored…”.
Line 281: Remove “conversely” and “ever”
Line 285-286: This should be placed also in the materials and methods section.
Line 288: Remove “roughly”. This term does not belong in a scientific article.
Line 291-294: This should be placed in the materials and methods section, not in the results.
Line 291: Replace “Most importantly, a protocol for the direct visual detection of LAMP results was developed” by “A protocol for direct visual detection of LAMP results was also developed…”.
Line 293-294: “The pH indicator color used turned from pink to yellow as a consequence of the pH value decrease during the LAMP reaction”.
Line 295-296: sentence not clear, please rephrase.
Line 297: Replace “were easily obtained as positive just after 15 minutes, or no longer than 30 minutes, on” by “were positive within a time period of 15 to 30 min on”
Line 299-301: What are the negative controls? In the article the negative controls seem to be water and non-infected cannellino plants and seeds? Any other negative control such as water with reagents only and plant/seed extracts, seed and plant materials from other plant species? What are the potential false positive/negative. Please explain. This sentences will need to be rephrased with more details.
Line 304-305: Visual detection of LAMP products obtained with bean seed samples cv. “Pak” for Cff detection.
Line 306: Replace “unifected” by “Uninfected”
Line 312: Replace “or” by “and”, or remove ”emerging or re-emerging”
Line 314-316: “The increasing worldwide spreading of Cff in bean‐producing areas is strongly driven by the global expansion of international trades of plant materials, and a direct consequence of climate change which impact on plant‐pathogen interaction has been demonstrated”. Need a reference for this sentence and the impact of climate change on plant-pathogen interaction.
Line 316: Replace “It would to be pointed out that” by “It is important to point out that”
Line 317-318: Replace “and help in fighting hunger in less developed countries, in addition to contribute to a healthy eating worldwide.” By “as they are a staple food and help fighting hunger in less developed countries and contribute to healthy eating habits worldwide.”
Line 319: Remove “even”
Line 322: Replace “much more deeper researches” with “more research”
Line 327-329: Replace “Conversely” with “indeed” and remove “basically”
Line 331-334: “LAMP has also several advantages in comparison to PCR‐based approaches, such as its low cost, easiness to use and suitability for in field/on site testing.”
Line 335-337: The only challenging step of LAMP approach is the need to correctly design effective primers. Because of the high number of primers needed for each primer set and the limited options as far as target sequences are concerned, the risk related to the formation of secondary structures can be high.
Line 338-340: In this paper, we present the development of a LAMP assay for the specific detection of Cff. The newly developed assay targets the unique and highly conserved 306 bp sequence, that was successfully used for the design of the PCR‐based assay for Cff and widely applied for in planta testing.
Line 344: Replace “whilst just 5 minutes were needed to detect 100 ng/reaction” with “with 5 minutes needed to detect 100 ng/reaction”.
Line 346: remove “ even”.
Line 348: Replace “by naked eye” with “through naked eyes”
Line 349: Replace “ and just after half an hour of isothermal sample incubation. ” with “30min after isothermal incubation of sample”.
Line 350: Replace “here used” with “used here”
Line 352: remove “here”
Line 354: Not just on bean seeds, but also on other plant materials.
Author Response
Please, see the attachment

Reviewer 2 Report
Dear Author, your manuscript on setting up an accurate diagnosis of Curtobacterium flaccumfaciens sp. Flaccumfaciens using LAMP is on importance as it will facilitate controlling this quarantine organism at the EU borders for incoming vegetable seeds and the producer can use it for controlling seed quality.
I found only one editing error:
Line165 … and then used directly used in for the…..
Author Response
Please, see the attachment

Round 2
Reviewer 1 Report
A few more minor English edits.
Line 21-24: “According to the most recent EU legislation, Cff is included among the quarantine pests not known to occur in the Union territory, and for which the phytosanitary inspection consists mainly on visual examination of imported bean seeds.”
Line 53: “….and often represent an essential income in rural areas.”
Line 55: Replace “planty” by “plant”.
Line 78-79: “…any C. flaccumfaciens isolate pathogenic, regardless of its host of 78 isolation.”…
Line 87: “…that are yellow…”
Line 89:”… flaccumfaciens isolates that are not pathogenic on bean, and the strains of the four other C. flaccumfaciens…”
Line 169: Remove 1 “used”.
Line 170-171: “…six samples for each condition (i.e. healthy-looking and symptomatic Cff-infected seeds, and certified 170 Cff free seeds) and each run were used.
Line 199: Replace “also because” by “as”.
Line 209: “…template with pure DNA…”
Line 214: “…gel in positive samples for all the tested amplification temperatures..”
Line 216: “has shown” or “showed”.
Line 240: “…Cmm) that are pathogenic on common bean and seed-borne..”. The sentence 237-240 is unclear, especially the end. Is the bacteria taxonomically related to cff pathogenic on common bean?
Line 283: “When the LAMP reaction…”
Line 285: “…as template…”
Line 300: “… of the incubation at 63°C, when DNA extracted from the Cff-naturally infected bean…”.
Line 303-306: Samples containing as template DNA from Cff-free certified seeds cv. “Pak” always scored negative. Controls with DNA template replaced with water also scored negative. At last, no false positives or negatives results were detected.
Line 321: “…strongly driven by the global expansion…”.
Line 334: Replace “Actually” by “Indeed”.
Line 357: “…this is the first time that a LAMP assay…”
Line 359-361: Suggestion for the last sentence:
This LAMP assay provides a reliable, specific and sensitive testing procedure, able to reveal the presence of Cff DNA on bean seeds and other plant materials, rapidly and with the naked eye, thus substantially reducing the risk of disease outbreaks due to early detection.
Author Response
Response to Reviewer 1 Comments
A few more minor English edits.
Point 1: Line 21-24: “According to the most recent EU legislation, Cff is included among the quarantine pests not known to occur in the Union territory, and for which the phytosanitary inspection consists mainly on visual examination of imported bean seeds.”.
Response 1: Thank for the suggestion, revised as requested.
Point 2: Line 53: “….and often represent an essential income in rural areas.”
Response 2: Thank for the suggestion, revised as requested.
Point 3: Line 55: Replace “planty” by “plant”.
Response 3: Sorry for the mistake here left during the previous revision, thank for the suggestion, revised as requested.
Point 4: Line 78-79: “…any C. flaccumfaciens isolate pathogenic, regardless of its host of 78 isolation.”
Response 4: Thank for the suggestion, but the sentence was not revised, to avoid any possibility of misunderstanding. To say “any C. flaccumfaciens isolate pathogenic” would also mean any isolate belonging to any of the C. flaccumfaciens pathovars. Conversely, the primer pair CffFOR2/CffREV4 is able to detect any Cff isolate/strain, that is able to cause disease on bean, although sometimes isolated also from other hosts than bean, such as tomato, pepper, etc.
Point 5: Line 87: “…that are yellow…”
Response 5: Thank for the suggestion, revised as requested.
Point 6: Line 89:”… flaccumfaciens isolates that are not pathogenic on bean, and the strains of the four other C. flaccumfaciens…”
Response 6: Thank for the suggestion, revised as requested.
Point 7: Line 169: Remove 1 “used”.
Response 7: Thank for the suggestion, revised as requested.
Point 8: Line 170-171: “…six samples for each condition (i.e. healthy-looking and symptomatic Cff-infected seeds, and certified 170 Cff free seeds) and each run were used.
Response 8: Thank for the suggestion, revised as requested.
Point 9: Line 199: Replace “also because” by “as”.
Response 9: Thank for the suggestion, revised as requested.
Point 10: Line 209: “…template with pure DNA…”
Response 10: Thank for the suggestion, the sentence has been revised.
Point 11: Line 214: “…gel in positive samples for all the tested amplification temperatures..”
Response 11: Thank for the suggestion, revised as requested.
Point 12: Line 216: “has shown” or “showed”.
Response 12: Sorry for the mistake left, thank for the suggestion, revised as requested.
Point 13: Line 240: “…Cmm) that are pathogenic on common bean and seed-borne..”. The sentence 237-240 is unclear, especially the end. Is the bacteria taxonomically related to cff pathogenic on common bean?
Response 13: Thank for the suggestion, the text 237-240 was partly rephrased, to make clearer that Cmm is closely taxonomically related to Cff, while those bacteria able to cause disease on bean and here used as non-target bacteria are Psp and Xap.
Point 14: Line 283: “When the LAMP reaction…”
Response 14: Thank for the suggestion, revised as requested.
Point 15: Line 285: “…as template…”.
Response 15: Thank for the suggestion, revised as requested.
Point 16: Line 300: “… of the incubation at 63°C, when DNA extracted from the Cff-naturally infected bean…”.
Response 16: Thank for the suggestion, revised as requested.
Point 17: Line 303-306: Samples containing as template DNA from Cff-free certified seeds cv. “Pak” always scored negative. Controls with DNA template replaced with water also scored negative. At last, no false positives or negatives results were detected.
Response 17: Thank for the suggestion, revised as requested.
Point 18: Line 321: “…strongly driven by the global expansion…”.
Response 18: Thank for the suggestion, revised as requested.
Point 19: Line 334: Replace “Actually” by “Indeed”.
Response 19: Thank for the suggestion, revised as requested.
Point 20: Line 357: “…this is the first time that a LAMP assay…”
Response 20: Thank for the suggestion, revised as requested.
Point 21: Line 359-361: Suggestion for the last sentence:
This LAMP assay provides a reliable, specific and sensitive testing procedure, able to reveal the presence of Cff DNA on bean seeds and other plant materials, rapidly and with the naked eye, thus substantially reducing the risk of disease outbreaks due to early detection.
Response 21: Thank you so much for the valuable suggestion, revised as requested.
Again, we appreciated all your useful comments to improve the clarity and the overall value of this manuscript.
We look forward to hearing from you regarding this last revised version of our manuscript, that we do hope now to be suitable for publication in Microrgamisms MDPI.